# Predictors of Psychological Distress among Post-Operative Cardiac Patients: A Narrative Review

**DOI:** 10.3390/healthcare11202721

**Published:** 2023-10-12

**Authors:** William D. McCann, Xiang-Yu Hou, Snezana Stolic, Michael J. Ireland

**Affiliations:** 1School of Psychology and Wellbeing, University of Southern Queensland, Ipswich, QLD 4305, Australia; michael.ireland@usq.edu.au; 2Poche Centre for Indigenous Health, The University of Queensland, Brisbane, QLD 4067, Australia; x.hou@uq.edu.au; 3School of Nursing and Midwifery, University of Southern Queensland, Ipswich, QLD 4305, Australia; snezana.stolic@usq.edu.au

**Keywords:** heart surgery, cardiovascular disease, anxiety, depression, psychological distress

## Abstract

Following surgery, over 50% of cardiac surgery patients report anxiety, stress and/or depression, with at least 10% meeting clinical diagnoses, which can persist for more than a year. Psychological distress predicts post-surgery health outcomes for cardiac patients. Therefore, post-operative distress represents a critical recovery challenge affecting both physical and psychological health. Despite some research identifying key personal, social, and health service correlates of patient distress, a review or synthesis of this evidence remains unavailable. Understanding these factors can facilitate the identification of high-risk patients, develop tailored support resources and interventions to support optimum recovery. This narrative review synthesises evidence from 39 studies that investigate personal, social, and health service predictors of post-surgery psychological distress among cardiac patients. The following factors predicted lower post-operative distress: participation in pre-operative education, cardiac rehabilitation, having a partner, happier marriages, increased physical activity, and greater social interaction. Conversely, increased pain and functional impairment predicted greater distress. The role of age, and sex in predicting distress is inconclusive. Understanding several factors is limited by the inability to carry out experimental manipulations for ethical reasons (e.g., pain). Future research would profit from addressing key methodological limitations and exploring the role of self-efficacy, pre-operative distress, and pre-operative physical activity. It is recommended that cardiac patients be educated pre-surgery and attend cardiac rehabilitation to decrease distress.

## 1. Background

Internationally, cardiovascular diseases (CVD) are the leading cause of death [1]. In 2019, 32% of all deaths worldwide (about 18 million people) were from CVD [2]. CVD entails various ischemic symptoms (obstruction in blood vessels that reduce or stop blood flow) associated with blockages of 50% or greater in one or more major arteries [3,4]. To treat patients with CVD, coronary artery bypass graft surgery (CABGS) can reduce all-cause mortality between 18% and 34%, reduce cardiac mortality by 30% to 39%, reduce cardiac hospitalisations by 27% to 45% at ten years post-surgery, and generally increase patient Quality of Life (QOL) [5,6]. However, recovery from cardiac surgery can be long and arduous, and existing evidence suggests that the most pressing health concern for cardiac patients following surgery is psychological distress [7]. For the purpose of this review, distress is defined broadly as encompassing states of elevated anxiety, depression, and stress.

CABGS patients have a high prevalence of distress both pre- and post-surgery. Prior to cardiac surgery, distress is elevated, with between 30% and 40% of patients meeting cut-offs for clinical psychological diagnoses [8]. Post-CABGS, up to 50% of patients experience depression, either suffering depressive symptoms or meeting diagnostic criteria for Major Depressive Disorder (MDD) [8,9,10]. These symptoms have been observed to persist for over a year and negatively affect a patient’s recovery in a variety of ways, including reducing motivation and energy [11]. Furthermore, about 10.1% of patients after CABGS experience Generalised Anxiety Disorder, which is more than double the population prevalence rate of four percent [12,13]. Finally, cardiac patients with high stress are considered ‘high risk’ for developing depressive symptoms and persistent anxiety after surgery [14,15]. 

Importantly, distress predicts post-surgery health outcomes for cardiac patients. For example, increased depression predicts slight increases in all-cause mortality and further cardiac incidents such as stroke and heart failure [16]. Generalised Anxiety Disorder is strongly associated with greater rates of morbidity and hospitalisation, with a 62 to 74% greater rate of further CVD events [17,18]. High stress among cardiovascular patients has also been associated with a 127% greater risk of further cardiac events and a 151% greater risk of mortality [15]. 

While the direction of causal influence between distress and CVD requires further clarification, it is plausible that this relationship is bidirectional and reciprocal such that elements of both potentially intensify each other [19,20]. For example, individuals with MDD have a 150% greater chance of developing CVD [21] and likewise, the prevalence of MDD doubles among people who experience a major cardiac event and is observed at a rate greater than 50% among patients who have undergone CABGS [9,10,11,22]. MDD is associated with two times greater risk of cardiac events, and two to three times increased risk of long-term mortality post-operation [8], and the presence of CVD increases the risk of distress even without surgery [23,24]. These potentially reciprocal causal interactions are critical for understanding the complicated link between cardiac and mental health.

Three meta-analyses representing a combined total of 57 studies provide further evidence of this interrelationship [25,26,27]. Relative to non-depressed participants, these meta-analyses reported an increase of 130% to 181% rate of future coronary disease incidents (e.g., hospitalisation and cardiac attacks) among distressed participants, and 122% rate of combined cardiac mortality and cardiac attacks among depressed participants. This potential reciprocal relationship further highlights the importance of understanding and addressing psychological distress following CABGS.

### Aims and Research Question

Importantly, post-surgical distress appears variable and modifiable, and research shows it varies as a function of multiple factors. These include moderate-to-strong effects for social (e.g., marital status, social interaction), moderate effects for behavioural (e.g., physical activity, fitness), and moderate effects for environmental factors (e.g., employment) [5,28,29,30]. Therefore, a thorough understanding of the relationship between patient’ distress and these factors is necessary for more effective patient support and optimised health outcomes. 

Therefore, the objective of this narrative literature review was to synthesise and critically evaluate studies of CVD patients that assessed the role of personal (e.g., age, and gender), social (e.g., social support), and health service (e.g., cardiac rehabilitation) factors in predicting depression, anxiety, and stress post-surgery. The following research question was posed: what factors have been identified in published peer-reviewed research as potentially influencing adult cardiac patient distress post-surgery? Specifically, what biographical, psychosocial, and health service factors are associated with higher psychological distress post-cardiac operation?

## 2. Methods

This review summarises peer-reviewed evidence published in the English language. Using Ebscohost Megafile Ultimate between January and May 2022, the following databases were searched: Academic Search Ultimate, APA PsycArticles, APA PsycInfo, CINAHL with Full Text, E-Journals, Psychology and Behavioral Sciences, Cochrane Reviews, Springer and Science Direct. The following search terms were used in various combinations and permutations:

Depression OR mental health OR mental illness OR anxiet* OR anxious OR stress* OR distress AND (cardiac patient OR cardiac patients) AND (postoperative OR post-operative OR after surgery OR post surgery OR following surgery). 

To ensure maximum sensitivity, the search terms did not target specific biographical, psychosocial, and health service factors but rather allowed these to emerge inductively from the studies reviewed. Furthermore, citation searching was used for all factors and has been identified in the Appendix A (location of hyperlink to Appendix A here).

Search ‘hits’ were imported into the Mendeley Desktop library. The lead author screened identified studies for eligibility using a multi-step process. Duplicate entries were first removed. Following this, a broad assessment of relevancy using study titles was carried out. Relevant study abstracts were then evaluated against inclusion and exclusion criteria. Finally, the full study texts were used to conduct a final evaluation against the inclusion and exclusion criteria. The process and results of screening studies for inclusion are depicted below (see Figure 1) using the Preferred Reporting Items for Systematic Reviews and Meta-Analyses diagram [31]. Seven additional studies were found via citation searching, with two subsequently excluded for recruiting non-cardiac patient participants. The 35 studies sourced through databases and the four studies sourced through citation searching comprise the final 39 studies retained for review.

### Inclusion and Exclusion Criteria

Inclusion criteria required study participants to be at least 18 years of age and were patients who have undergone open-heart surgery. Studies were required to examine distress and one or more personal, social, and/or health service factor. Studies that investigated pharmaceutical interventions (e.g., anaesthesia, beta-blockers, ketamine, opioids, etc.) were excluded in the screening process in favour of behavioural-, social-, and cognitive-based interventions and factors. Pharmacology was considered outside the first author’s discipline of behavioural sciences and therefore deemed beyond the scope of this review (psychosocial and behavioural correlates). The studies reviewed also had the same scope as this review, primarily focusing on these behavioural, social, and cognitive factors as opposed to pharmaceuticals. Studies were required to be peer-reviewed and published in journals indexed in the databased listed above. No publication date restrictions were imposed. 

It should be noted that the current review was not intended to be comprehensive but instead to be succinct; therefore, systematic reviews and meta-analyses were considered as primary studies and included where possible. RCTs were considered secondary but were included in the review based upon the originality of their investigations or findings that would enhance the understanding of a given factor (e.g., contrasting findings to reviews, additional findings relating to distress, or if a study was not included in a review). To the best of the authors’ ability, the most current systematic reviews and meta-analyses were included in favour of older reviews for each factor.

## 3. Results

Thirty-nine studies from 1983 to 2022 were reviewed; these include approximately 200,000 participants across 16 countries and consist of qualitative, quantitative and review methodologies. The types of factors examined as predictors of distress among cardiac patients are grouped under the following themes: personal, social, and health service factors. The general effect sizes and trends for each specific are summarised in Table 1.

Four personal factors have been the subject of research: age, sex, pain and functional impairment, and physical activity. A systematic review found mixed evidence for the role of age as a predictor of post-operative distress [5]. McKenzie and colleagues [5] reviewed 46 studies published between 1960 and 2008, with the focus on reviewing pre-operative predictors of post-operative distress in CABGS patients. Five studies reviewed reported small-to-moderate associations between younger age and elevated distress (e.g., Oxman et al. [32], Okkonen and Vanhanen [33], etc.), three studies reported small associations between older age and elevated distress, and five studies reported no association (e.g., Caccamo et al. [28], etc.). McKenzie and colleagues [5] reasoned that these mixed results are due to many of the studies comparing age groups that were not sufficiently different. For example, in one study where age did not predict distress, the ‘older’ group had a mean age of 78 years compared to the ‘younger’ group mean of 65 years [34]. In another study where younger age was weakly associated with higher distress, patients were all aged between 61 and 81 years [35]. In these cases of limited variance in participant age, the predictive value of age may be attenuated. However, a simple bivariate analysis of linear differences across dichotomised age groups may be an inadequate way to study the link between age and distress. 

There are several factors that may moderate the association between distress and age. For example, there is evidence of increased post-operative complications associated with older age; however, greater utilisation of social resources, such as involvement in religious groups, charitable organisations or sporting clubs, decreases these post-operative impacts [36,37,38,39]. Additionally, trait resilience may moderate a potential association between age and post-operative distress. Older CABGS patients (aged between 55 and 75 years) self-reported much higher resilience three months post-surgery than younger patients (aged between 30 and 45 years) [40]. Sanyal and colleagues [40] argued that this may be due to the different coping skills and styles older individuals attain with age. This is consistent with a pattern of greater resilience observed in older individuals among the general and other specific populations [41]. Notably, Matzka and colleagues [42] reported that among cancer patients, age moderated the association between distress and resilience, such that older age was moderately associated with lower distress as a function of greater resilience. With respect to the research question, these findings suggest that age potentially influences cardiac patient distress post-surgery with small-to-moderate increases in distress among younger patients, and small increases among older patients.

The available evidence is also unclear as to whether the cardiac patient’s sex leads to a difference in post-operative outcomes. McKenzie and colleagues’ [5] systematic review reported that sex only predicted depression in 72% of reviewed studies. Similarly, sex predicted anxiety in 50% of reviewed studies. Studies that reported an association between sex and distress consistently reported small-to-moderate higher distress amongst female participants. The authors note that in many studies where sex was not found to predict outcomes, female participation was limited. Therefore, the extent that sex predicts post-operative distress remains unclear; however, its role in predicting post-operative depression remains plausible but requires further corroboration.

Evidence relating to other trends could shed light on why and how sex may relate to distress among cardiac patients. Female patients self-report greater severity of chronic pain, physical impairment, and long-term post-CABGS pain [43,44]. Since pain and physical impairment are known risk factors for distress among cardiac patients, it is likely that females might be at greater risk of future distress [45]. Furthermore, females appear generally more likely to report distress in the population. For example, they are twice as likely as males to be diagnosed with depression [46]. Whilst the available evidence is in conclusive as to whether sex predicts distress, there are apparent susceptibilities such as greater general female disposition for distress and evidence of long-term complications (mostly involving pain) following surgery. With respect to the research question, these findings suggest that the sex of a cardiac patient potentially influences their distress post-surgery with females having small to moderate increases in distress compared to males. 

After open-heart surgery, many patients experience severe chest pain or discomfort called angina, which in turn can generate and exacerbate distress. Angina occurs when blood travel is restricted through the vessels of the heart causing a reduction in oxygenation [47]. Gravely-Witte and colleagues [48] carried out a six-month longitudinal survey and reported that cardiac patients who experienced symptoms of angina had small increases in depression and small reductions in emotional QOL compared to patients without angina. 

Similar to the bidirectional relationship between distress and CVD, pain acts as both a symptom and an aggravator [49]. Specifically, increases in pain can reduce adherence to prescribed medical treatments, which in turn decreases QOL, increases symptom burden, fosters greater dependency on health care services, increases functional and role impairment, and increases the risk of depression among post-operative cardiac patients [7,50,51,52,53,54]. Longitudinal research highlights the bidirectional link between pain and elevated distress. 

Perski and colleagues [55] conducted one of the earliest three-year longitudinal surveys of CABGS patients. Prior to surgery, participants who were distressed reported moderate increases in self-reported pain compared to those who were not distressed. This trend was also apparent one-year post-surgery though the difference in distress was small. These findings are consistent with research on patients undergoing non-cardiac surgeries, which report that reduced pain thresholds exist amongst distressed individuals [56,57]. 

A more recent RCT by Morone and colleagues [58] reported that at 12 months post-hospitalisation, participants with greater depression had large increases in pain scores compared with non-depressed participants. From four to twelve months, patients reported that their pain remained relatively unchanged and non-depressed patients still had elevated levels of pain. Although all patients reported a decrease in depression at 12 months, participants with moderate or higher self-reported pain demonstrated much higher depression compared to participants with low or no pain. From this evidence, it is clear that following surgery, pain and distress co-vary to a small degree among CVD patients; thus, in answer to our research question, pain does predict slight increases in distress post-surgery. 

Physical activity appears to be an important factor mitigating distress for cardiac patients following surgery. Patients with greater physical fitness are also less likely to experience CVD risk factors and future cardiac events [59]. Low physical activity is also a risk factor for distress among patients both pre- and post-surgery [60,61]. 

A meta-analysis showed that among those with chronic illness, including cardiac patients, greater physical activity was associated with a small decrease in anxiety and depression [62]. Herring and colleagues [62] reviewed 90 studies published between 1998 and 2011, with the focus on reviewing the effect of exercise training on depression among patients with chronic illnesses. They analysed 168 effects from those studies with 23% being from patients with CVD. The findings were mixed with the association between physical activity and distress dependent on patients either having pre-existing depression or post-operative emerging depression. Among cardiac patients with post-operative depression, the lack of self-reported physical activity during six months post-surgery was associated with over double the risk of increased depression [63]. In comparison, patients who had pre-operative depression had their symptoms peak at the time of hospitalisation, and then gradually decrease during the following six months. This trend of small reductions in pre- to post-operative depression has also been reported by studies that did not investigate physical activity (e.g., Linden et al. [64] and Hussain and Moeen [65]). 

Among populations with serious mental health disorders (for example, patients with MDD), the odds of meeting ‘sufficient’ physical activity levels (as stipulated by international guidelines) [66] is 150% lower than the general population [67]. This means that post-surgery, the 50% of patients that experience highly elevated distress are at risk of sedentary behaviour [10]. However, with increased physical activity post-surgery among patients with MDD, there are large increases in positive mood, enhanced well-being, and higher QOL compared to patients that did not receive physical activity interventions [68,69,70]. For instance, Herring and colleagues’ [62] review of RCTs reported that exercise training amongst patients with chronic illnesses reduced anxiety, fatigue, pain, and improved QOL with similar effect sizes to pharmacotherapy. Therefore, physical activity shows a reliable association with distress, and patients that are clinically depressed potentially reap the greatest benefit.

In a prospective cohort study, Kehler and colleagues [71] investigated pre-operative physical activity among cardiac surgery patients and its impact on post-operative depression and complications such as rehospitalisation, length of stay at hospital and mortality. They reported that post-operative depression was not independently associated with physical activity. However, it was identified that post-operative complications may be a confounding variable between distress and pre-operative physical activity. It was reported that depressed patients who had high pre-surgery activity had 35% lower risk of post-operative complications compared with depressed patients who had low pre-surgery activity. These results suggest that pre-operative physical activity may be an important predictor of depression. With respect to our research question, these findings suggest that physical activity predicts slightly lower cardiac patient distress post-surgery.

Among the social factors that have been investigated are family and partner relationships, and social interaction and social activity. The reviewed evidence indicates a relationship between marital status and distress for post-operative CVD patients. Kaplan and Kronick [72] used National Health data from the USA to investigate the relationship between marital status and mortality among a mixed sample of CVD patients and the general population. The results showed that never-married participants had a small increased risk of CVD mortality compared to married participants. This result may have been explained by the slight increase in smoking behaviour and alcohol consumption observed between married and unmarried groups. A recent prospective study by Caccamo and colleagues [28] investigated the impact of cardiac rehabilitation on social inhibition and distress among post-operative cardiac patients. The results showed that participants with a partner reported much less anxiety and stress, compared to patients without a partner. 

Furthermore, Dhindsa and colleagues’ [73] review of 14 large-scale studies (from 1993 to 2018), most including over 10,000 participants each, concluded that, among unmarried CVD patients, there is a slightly higher prevalence of distress. Unmarried CVD patients were also found to be twice as poor with medication adherence. Furthermore, patients with carers or partners reported increased participation in cardiac rehabilitation programs, medication, diet, and exercise adherence, and reduced hospital readmissions compared to patients without a partner or carer [73]. These authors argued that the quality of marriage would be an important consideration for future studies as unhappy marriages have worse outcomes with regard to general QOL and distress than happy marriages in both the general population and among CVD patients [73,74]. 

The quality of marriage among post-operative CABGS patients was examined in an early interview study, where it was found that the majority (81%) of the 318 patients reported receiving sufficient affection from their family and marital relationships six months post-surgery, with half reporting that their family relationships had grown closer (only three percent reported that their martial and family relationships grew further apart as a result of their surgery) [75]. Although a majority of patients in this study reported ‘sufficient’ affection, a sizeable portion of patients had a negative or no increase in marriage quality, which warrants underscores the need for further investigation into the relationship between marriage quality and distress in post-surgery patients.

De Fazio and colleagues [76] investigated Type D personality (the tendency towards social inhibition) and distress among post-surgical cardiac patients and reported elevated symptoms of distress in over three-quarters of all participants. The results showed that greater distress was associated with Type D personality, unmarried participants, and coping strategies that included social support seeking. Specifically, post-surgical cardiac patients with Type D personality had large increased odds of greater anxiety and depression, and unmarried participants had vastly increased odds of greater anxiety. Participants high in social-oriented coping (e.g., engaging in social activity to reduce distress) had large increased odds of greater anxiety post-surgery. The authors argued that greater anxiety reported among unmarried patients may be due to the fear of being alone if another cardiac event occurs, which is a common concern among cardiac patients [29]. Unmarried cardiac patients have been reported to become increasingly socially isolated, reducing the number of emotionally close people in their lives [32,77]. However, the findings of greater anxiety in participants with high social oriented coping highlights the potential for heavily investing in social activities to impose some emotional cost. Regarding the research question, these findings suggest that family and partner relationships predict small to large decreased cardiac patient distress post-surgery.

Following surgery, the social lives of cardiac patients appear largely maintained or expanded. Compared to one to three days before surgery, Jenkins [75] reported 42% of participants interviewed 6 months post-surgery reported increased social activity (e.g., self-initiated activities, etc.), 40% maintained a steady pattern and 18% reported a decrease. Moreover, for social interactions (e.g., talking to friends, relatives and others face-to-face), 48% of participants reported a similar degree of interaction both before and after their operation, whilst 28% reported an increase and 24% reported a decrease. Further studies have corroborated this trend for stable or increasing social engagement after cardiac operations (Linden et al. [45] and Nielsen et al. [46]) with these interactions occurring through participation in religious organisations, community groups, and close relationship circles. 

Greater social support appears to play a protective role in relation to psychological, cardiac and physical health. For example, a prospective survey study examining social and religious activity during the six months post-cardiac surgery reported that a lack of social participation was associated with a three times greater risk of mortality during this period [32]. Furthermore, a lack of social participation was slightly associated with greater depression, higher neuroticism, and lower extraversion. This suggests that patients who are depressed and/or with higher neuroticism are less likely to become involved in social activities and are at considerably greater risk of cardiac mortality in the months following surgery. 

Carers of cardiac patients often provide considerable physical and emotional support once the patient leaves hospital [78]. Having carer support is associated with a moderate reduction in post-operative distress, and this trend has been found across reviews [79], and quantitative studies [80,81,82,83,84]. However, this association with decreased distress may be confounded with the tendency for distress to decrease over time after cardiac operations [30,64,65]. With respect to our research question, these findings suggest that social interaction predicts moderately lower cardiac patient distress post-surgery.

Among the health service factors that have been investigated are pre-operational education and cardiac rehabilitation. The available evidence shows that pre-operative education is linked with improvements in patients’ physical health, improved QOL, and reduced distress [85,86]. Both pre-operative education and cardiac rehabilitation were the most well-studied factors with six studies found for each factor. 

The following describes evidence relating to the effect of pre-operative education on distress from the most rigorous studies (i.e., RCTs are the only design from which causality can be inferred). Sørlie and colleagues’ [87] RCT assigned participants at pre-surgery to an intervention in which they attended two in-person (e.g., specifically 40 min nurse-run) sessions and two video information sessions at admission and discharge. The training emphasised trusting relationships, encouragement to express feelings, learning to confront difficult and emotional situations, and motivation to seek information. Control participants received standardised information by nurses at admission and discharge (e.g., procedural and behavioural instructions). Intervention participants had exposure to experienced staff who were advised to tailor their approach to suit individual patients. Compared to those in the control group, intervention participants had moderately lower anxiety at discharge and at one year post-surgery. Furthermore, the intervention group had moderate improvements in depression at the six-month and two-year follow-up assessments. The authors argued that since nurses were overseeing the tailored intervention sessions, the interaction was greater between participants and nurses than it would have been in the control group. Other research has shown that when nurses encourage questions and are able to provide individual feedback and support, patients’ anxiety decreases [88]. Therefore, it is likely that both pre-operative education and greater health staff interaction were responsible for observed improvements in anxiety.

Hoseini and colleagues’ [89] RCT investigated a post-surgery intervention in which all participants received routine training (e.g., general information on post-operative milestones and potential complications), however, some were randomised to receive additional audiotaped education (e.g., providing training and information on post-operative home care) before discharge. Compared to control participants, large reductions in anxiety and depression were reported post-intervention by education intervention participants and these were maintained at six weeks post-surgery. This is of note given intervention participants had moderately higher depression pre-surgery compared to controls. 

Shelley and Pakenham’s [90] single-blind RCT identified small improvements in patient’s distress; however, these were moderated by the patient’s belief in the medical professionals and their self-efficacy towards improving their health. Prior to CABGS, all participants in this study received usual care with visitations by a doctor and anaesthetist to discuss the procedure (standardised information) and answer questions; however, intervention participants were assigned to also receive one-on-one additional education and coping training by a psychologist (e.g., cognitive coping strategies). Participants in the intervention group who had high general self-efficacy and an external health locus of control (belief that outcomes are the result of others such as medical professionals, family and friends) reported slightly lower distress and lower pain relative to control participants. However, among intervention patients, having a high external health locus of control and low self-efficacy were associated with an increase in distress. Therefore, patients believing strongly in medical staff and not in their own ability to control their recovery appear to be worse off or, at least, unaided by the pre-operative preparation compared to controls. 

Guo and colleagues [91] conducted an RCT of a pre-operative education intervention in which participants were given information leaflets by a cardiac nurse. Compared to a no-education control group, at one-week post-surgery, the education group reported moderate reductions in depression and a large reduction in anxiety. Interestingly, the education group also spent less hours in the cardiac surgical intensive care unit, but overall length of hospital stay was not different. 

In Asilioglu and Celik’s [92] two-group non-randomised comparative study, an intervention group was educated on information about their surgery, heart disease, and medicine by a nurse using a structured education booklet. Members of the control group were matched with intervention participants on age, sex, marital status, education level, employment, social and family support, and history of anaesthesia. The control group received standard care (standardised information). No significant differences in distress were reported between the two groups at three days post-surgery. However, during this time, patients are generally still in the intensive care unit on a range of medications for pain and may suffer reduced cognitive ability with affected attention, disorganised thinking, fluctuating consciousness, and mental status [93]. Participants at this timepoint may not have been able to accurately reflect, or report, on their psychological state [94]. 

The merit of pre-operative education has been investigated in a meta-analysis by Ng and colleagues [94]. They reviewed 22 trials (3167 participants) and reported that pre-operative education was associated with large reductions in time spent in the intensive care unit, small reductions in post-operative depression, and increasing patient satisfaction with care. Additionally, compared with usual care, pre-operative education was associated with large reductions in pre-operative anxiety among patients. The positive effect of pre-operative education was greater in studies where patients only underwent CABGS compared to simultaneously CABG and other open-heart surgeries. The results relating to post-operative distress reported by Ng and colleagues [94] overlap with two studies presented above both being Guo and colleagues [91] and Sørlie and colleagues [87]. These two studies were discussed separately from the review by Ng and colleagues [94] as they had additional findings of interest, those being time spent in hospital and the possible effect of nurse interaction, respectively. Regarding the research question, these findings suggest that pre-operative education predicts lower cardiac patient distress post-surgery.

Patients who attend cardiac rehabilitation generally undergo a series of activities including education on healthy behaviours (smoking, diet, physical activity) and distress [95]. These programs are designed to assist patients to recommence day-to-day living following surgery. However, there are additional elements to cardiac rehabilitation that may influence patient wellbeing, such as group-led sessions, patient-perceived quality of rehabilitation information, and psychological care. Generally, favourable outcomes have been observed following cardiac rehabilitation, with evidence showing small reductions in post-surgery cardiac events, and small-to-moderate reductions in distress post-surgery [96,97]. 

Despite these positive outcomes, several studies suggest cardiac rehabilitation is not a panacea for resolving post-operative problems. For example, in a prospective study by Caccamo and colleagues [28], following two weeks of cardiac rehabilitation, patients had a small increase in anxiety compared to pre-cardiac rehabilitation. The authors speculate that this may have been due to clinical education and group-led sessions increasing patient awareness of risks and potential problems. This study also reported moderately higher depression scores for unemployed versus employed participants and the authors postulated that unemployed participants may have had increased time and capacity for introspection as seen in other studies on distress among unemployed people [98,99]. An encouragement of introspection during cardiac rehabilitation may have also exacerbated this increased depression. 

The perceived quality of rehabilitation may also influence patient distress during recovery. Davies [78] surveyed cardiac patients and carers at one and seven weeks post-discharge about the quality of information they received regarding their recovery. Patients who perceived ‘high’-quality information reported moderately lower anxiety and relatively stronger reductions in depression compared with those who perceived ‘lower’-quality information at seven weeks post-surgery. Therefore, high-quality cardiac rehabilitation appears to have a positive association with reduced patient distress post-surgery. 

The available evidence suggests that a component of cardiac rehabilitation, psychological treatment, is effective at reducing depression and stress but not anxiety. A meta-analysis of 23 RCTs reported that psychological treatment was associated with a small increase in perceived social support and QOL compared to control groups [64]. These effects also appear to vary as a function of gender. Specifically, among men, reductions in depression and improved perceived social support were small. However, among women, while small reductions in depression were also observed, improvements in perceived social support were moderate. Regardless of gender, post-surgery anxiety was not different between intervention and control conditions. This may indicate that the normal trend of diminishing anxiety in the period following surgery (see Okamoto and Motomura [100] and Sharif et al. [101]) is not altered by psychological treatment.

Linden and colleagues’ [64] meta-analysis reported that the timing of cardiac rehabilitation post-surgery may be associated with changes in distress. Studies that initiated cardiac rehabilitation soon (<two months) post-surgery reported little psychological and mortality benefits compared to usual-care control groups. However, the effects on mortality appear to be associated with distress. Specifically, when distress was high or unchanged, cardiac rehabilitation interventions yielded a 13% increase in mortality but when distress was reduced, rehabilitation interventions yielded a 31% decrease in mortality. Therefore, it is plausible that initiating rehabilitation later may not only decrease mortality but also distress, although the reference studies did not investigate the timing of rehabilitation and the outcome of distress. With respect to our research question, these findings suggest that attendance of cardiac rehabilitation is a mixed predictor of cardiac patient distress post-surgery with small, moderate, and large decreases in distress but also some reports of small to moderate increases in distress.

## 4. Discussion

The aim of this review was to synthesise and evaluate the available evidence on the role of biographical, psychosocial and health service factors in predicting distress among cardiac patients post-surgery. This review uncovered a greater body of research and has investigated the health service factors (12 studies) predicting post-operative distress compared to physical (11 studies) and social factors (six studies; see Table 1). The evidence reviewed here suggests that pre-operative education, family and partner relationships, cardiac rehabilitation, physical activity, and social interaction reduced distress post-surgery. Conversely, the evidence suggests that increased pain and functional impairment predicted greater distress post-surgery. Despite being the subject of multiple studies, the available evidence remains inconclusive regarding the influence of physical factors such as patient age and sex. 

There are varied findings regarding the ability of age to predict patient distress and cardiac outcomes. It appears that older patients have less access to, and diversity of, social resources, but benefit from greater personal resilience, which may explain the mixed results with resilience offsetting social resource deficits. Higher resilience in other populations (e.g., medical and psychology students, and cancer patients) has been associated with lower distress [42,102]. Therefore, resilience among older post-operative cardiac patients may be an important protective factor that buffers against distress. However, a cross-sectional study of 178 CABGS patients found there were small increases of pre-operative anxiety with increased age; however, increased age did not impact levels of pre-operative depression [103]. This study highlights that age alone may not predict all aspects of distress. Furthermore, for studies investigating the relationship of age and distress, there appears to be three limitations. First, studies appear to categorise the continuous variable of age. Two studies have apparently only looked for linear effects, yet the effect may be non-linear as it appears that sometimes older age relates to greater distress and other times lower distress. Therefore, future studies should measure age as a continuous variable and test for non-linear relationships between age and distress. 

The available evidence is also inconclusive as to whether sex predicts distress following cardiac surgery. Upon further scrutiny, 8.7% of studies reviewed by McKenzie and colleagues [5] used depression scales that are susceptible to being confounded by somatic symptoms (e.g., Symptom Checklist-90). This may have altered how each sex reported depression given that females report depressive symptoms connected to their physical states at a higher rate compared to males and contributed to the overall inconclusive findings [104]. Furthermore, females are generally more prone to reporting greater discomfort, long-term pain, and greater general disposition to distress [43,44,46]. It is clear that further research is needed to explore the differences between sex on psychological outcomes using samples that have at least a similar portion of female participants to males due to the noted lack of female participation in the reviewed studies.

It is clear from the reviewed evidence that self-reported physical pain is associated with post-surgery distress among patients. Increases in pain were variously associated with small and large increases in distress post-surgery [48,55,58]. It appears that the association between physical pain and distress is a bidirectional relationship, which is not a phenomenon that is unique to cardiac patients given physical pain is also associated with exacerbating mental health disorders, osteoarthritis, and additional chronic conditions [49,105,106]. Interestingly, distress appears to be comorbid with pain for most of these illnesses [106]. 

The evidence is also clear that physical activity influences distress post-surgery. Patients who engage in greater physical activity have small to moderate reductions in distress post-surgery with deficient activity being associated with two to three times greater rates of depression [63]. Interestingly, increased physical activity post-surgery appears to be most beneficial for patients who developed high distress post-surgery compared to those with high pre-operative distress. Patients with MDD had a 150% lower rate of physical activity, supporting the notion that greater distress is associated with less physical activity [66]. Furthermore, increased pre-surgery levels of physical activity were associated with less post-surgery distress, which is an important finding as many studies only focus on investigating post-operative physical activity [71]. Future investigations should continue to investigate pre-operative physical activity to better understand the strength of this effect on distress. 

The reviewed evidence suggests that family and partner relationships are associated with reduced patient distress following surgery. Patients with a partner appear to reliably report at least small reductions in distress, with large reductions in distress post-surgery also observed [28,73,76]. It is apparent that close social support (e.g., family, carers and partners) reduces distress and therefore increases the ability of patients to cope during their recovery from surgery. Further exploration of marriage and partner relationships is required as the quality of these relationships is an important facet of recovery for cardiac patients post-surgery [73]. Due to a sizeable portion (11.3%) of cardiac patients reportedly living alone, and the established effect of family and partner relationships on post-operative distress, these both warrant further investigations and attention from medical practitioners to reduce the negative effect on distress and other health outcomes [107]. Based on these findings, there is little doubt that access to family and partner relationships is an important protective factor shaping patient health, social support and distress.

The majority of reviewed studies of social interaction reported patient benefits with small to moderate reductions in distress. Improvements in both QOL scores and distress were reported among patients who were regularly involved in social group events, and greater social support appears to decrease the impact and severity of distress among patients [32,78]. This finding is reflected in the results of a prospective survey study that investigated social interaction and depression among 68 CABGS patients [108]. After one week of surgery, patients had large increases in self-reported depression and a minimal increase in self-reported social interaction. However, at three and six months post-surgery, there were large reductions in depression and large increases in social interaction. In another prospective survey study of anxiety and depression among patients with recent acute cardiac events (>50% CABGS) [109], Murphy and colleagues [109] found that reduced social interaction (unpartnered and/or living alone) predicted greater distress at two to four months and six to twelve months post-surgery. Patients that were living alone had small increased odds of depression and small increased odds of having both anxiety, and depression at six to twelve months post-surgery. Unpartnered patients had small increased odds of being anxious, depressed, and comorbidity. It was reported by Xia and Li [110] in their review of molecular mechanisms that social isolation impacts the autonomic nervous system, reducing cognitive function, increasing the risk of cardiovascular disease, and depression. Therefore, these data highlight the importance of patients having sufficient social interaction as low levels of interaction are associated with adverse health outcomes. Further investigation among post-operative cardiac populations would be beneficial to determine if social interaction is an independent predictor of distress and to disentangle the effect of social interaction from the normal improvement in distress over time. 

From the evidence reviewed, pre-operative education appears to influence patients’ post-operation recovery in several important ways. The majority of studies reviewed reported small to moderate associations between attending pre-operative education and decreased patient distress from one-week to one-year post-surgery. However, up to six weeks post-surgery, large reductions in distress were reported [89,91]. In the instance where no associations between pre-operative education and distress were reported [92], the authors measured patient distress three days post-surgery, which is considered unusually early for accurate reflections of distress [94]. Studies investigating additional parameters, such as self-efficacy and locus of control, demonstrate that the influence of pre-operative education may be moderated by dispositional characteristics, and this could explain why some participants in certain studies did not show expected improvements [90]. Future studies might benefit from measuring other dispositional characteristics when implementing educational interventions as this will shed greater light on the pattern of findings. Pre-operative education appears most effective in reducing patient anxiety from pre- to post-operation relative to usual care and, to a smaller degree, reducing post-operative depression. However, it should also be noted that pre-operative education did not exhibit effects on pain, future cardiac complications, total length of hospitalisation or QOL [94]. The absence of an effect across measures other than distress is interesting given the established associations between QOL and anxiety, depression, and stress [87]. Therefore, these trends warrant further investigation. 

Participation in cardiac rehabilitation appears to reliably produce at least small reductions in distress, although moderate-to-large reductions in distress post-surgery have also been observed [78,97]. It should be noted that large reductions were observed when rehabilitation information was perceived as high quality by patients [78]. Furthermore, one study found rehabilitation was associated with small to moderate increases in distress [28]. It is noted that in this study, that a longer period of assessment may have allowed for a better determination of participant distress as measurement occurred post-surgery at the start of rehabilitation and at two weeks post-surgery (end of rehabilitation). More commonly, cardiac rehabilitation lasts on average 13 weeks and therefore this study may not reflect routine rehabilitation effects [97]. Several elements of cardiac rehabilitation were identified as responsible for positive effects, such as group-led sessions, information quality, and psychological treatment. However, it is apparent that important moderators of these effects (e.g., timing of rehabilitation, perceived quality of information, etc.) have received insufficient attention to date [64,78]. Finally, extra care should be taken when encouraging patient introspection in order to prevent negative rumination that may occur [28,90]. Despite these positive findings, low participation rates remain a major issue for health practitioners and researchers. Specifically, with estimates that only 30% of people eligible for cardiac rehabilitation participate, the majority are not able to benefit [111]. 

For factors such as pain and functional impairment, and physical activity, there appears to exist a bidirectional relationship with distress. This poses additional challenges for patient recovery, keeping a patient in a self-perpetuating loop that is detrimental to their health. This also further underscores the importance of medical practitioners and researchers attending to these concerns. Although several factors (e.g., adherence to medication, etc.) appear to moderate the relationship between distress and both physical activity and pain, and functional impairment, there have been only limited investigations exploring this bidirectional relationship. Therefore, it is unclear whether and how this bidirectional cycle can be broken, or at least, reduced in severity.

The evidence reviewed suggests that resilience and self-efficacy appear to influence both age and cardiac rehabilitation on distress. It was reported that, with greater self-efficacy, patients experienced moderate reductions in post-operative distress [90]. Therefore, pre-operative education appears most advantageous (at least for distress reduction) among patients with greater self-efficacy. The effect of incorporating strategies to boost self-efficacy in future pre-operative interventions would be a useful avenue for future research [90]. Additionally, screening for self-efficacy prior to surgery would allow the identification of patients that might require additional support. In addition to self-efficacy, other personal characteristics such as dispositional resilience and age warrant further attention. These two factors may not be mutually exclusive as older CABGS patients report greater resilience than younger patients from admission to three months post-surgery [40]. Furthermore, the result of greater resilience amongst older cardiac patients appears to produce moderate reductions in post-operative distress [42]. 

This reviewed studies recruited patients from 16 countries including Australia [90], Canada [48], England [5], Germany [112], India [40], Iran [101], Italy [76], Japan [100], Malaysia [33], Netherlands [34], Norway [43], Singapore [29], Spain [59], Sweden [55], Turkey [92], and the United States of America [65]. From such a wide array of origins, it is possible that cultural factors may have influenced the relationship between distress and demographic factors to some degree. For example, Guo [113] noted that non-Western countries (e.g., China, India, etc.) do not routinely prioritise the provision of health information to patients in their healthcare systems. While receiving little information may cause additional anxiety in patients in Western countries (who might expect to receive such information) this might cause no additional concern for patients in non-Western countries. Therefore, studies exploring greater pre-operative education with Western patients may observe greater improvements in distress than studies with non-Western patients.

Two limitations of this review are worth noting. First, there was a possibility that relevant studies were not covered by the database search and therefore were not included in this review. The search methodology aimed to identify as many relevant studies as possible without including many irrelevant studies; that is, it emphasised specificity over sensitivity. Second, to keep this review concise, a comprehensive review of all factors predicting distress and a complete review of each factor was not performed. Instead, upon reviewing studies that met inclusion criteria, pharmaceutical interventions (anaesthesia, beta-blockers, etc.) were excluded in favour of personal, social, and health service factors. This was carried out as pharmacology was considered outside the first author’s discipline of behavioural sciences and therefore deemed beyond the scope of the review.

### 4.1. Implications for Future Research

This review highlights potential implications for future research into the trajectory of psychological health of CABGS and valve surgery patients. Methodological challenges prohibit experimental research into many factors reviewed here; specifically, factors like sex and age are not experimentally modifiable, whereas other factors, such as pain and functional impairment and social interaction cannot be manipulated due to ethical concerns. Therefore, it is difficult to empirically disentangle the role of these factors from the normal trajectory of diminishing distress over time post-surgery [58,101]. Future research would benefit from having more comprehensive measurements of marriage, relationships or social interactions (e.g., those also assess their quality), having greater female representation when investigating patient sex, and assessing age as a continuous predictor of distress, while also exploring the potential for nonlinear relationships among these factors. 

When investigating the influence of interventions such as pre-operative education and cardiac rehabilitation (which can vary considerably in their content), it will be fruitful to incorporate the investigation of key mediators and moderators of observed effects such as different formats (e.g., via telehealth services, etc.), and program structures (e.g., length of rehabilitation, etc.). For instance, it was reported that the timing of cardiac rehabilitation (e.g., two weeks post-surgery compared to two months post-surgery, etc.) had different effects on post-operative distress, and that delayed cardiac rehabilitation (two months post-discharge) had a positive influence in reducing distress compared to the standard timing of rehabilitation [64]. Medical professionals and researchers should examine and attempt to explain variance in post-operative factors such as participation in cardiac rehabilitation, levels of physical activity, and social activity when treating cardiac patients after surgery as these activities are associated with reductions in distress. Measuring self-efficacy would provide a more accurate interpretation of the relationship between both cardiac rehabilitation and pre-operative education on post-operative distress. For example, a study may find no effect of cardiac rehabilitation on distress; however, these results might be owing to many low self-efficacy patients being included in the sample, and that amongst the high self-efficacy patients, there was a strong effect. This is also true for measuring the timing of cardiac rehabilitation when investigating the relationship between cardiac rehabilitation and distress. Moreover, measuring pre-operative levels of physical activity would benefit investigations between post-operative physical activity levels and distress. 

Furthermore, this review found that participation in pre-operative education consistently predicted decreased distress but did not have an effect for measures such as quality of life, pain, future cardiac complications, and length of hospitalisation. This is unexpected given distress and these interrelated outcomes are empirically associated with each other [58,64,87]. Future research may find merit in investigating why there is an absence of effect across other measures for pre-operative education. Future research questions should also pertain to cardiac rehabilitation given the minimal research into which elements of cardiac rehabilitation (e.g., nutritional counselling, exercise training, etc.) are the most effective in reducing distress and the optimal timing of rehabilitation programs [64]. This could lead to better tailored rehabilitation programs which may reduce time required for patients to participate, thus making it both more feasible and attractive to patients. Regarding cardiac rehabilitation, future research should also attempt to uncover potential barriers to participation and test interventions to mitigate these. Finally, biographical, cultural and health system contexts present as important avenues for future research. It remains unclear how cultural and demographic variables interact with other factors to shape post-operative distress.

### 4.2. Clinical Implications

Several clinical implications are apparent from the evidence reviewed including some that can be implemented into normal practice for hospitals and medical professionals both before and after surgery. Given family and partner relationships are a protective factor for distress post-surgery, it is clear that at least 11% of cardiac patients are at greater risk due to living alone [107]. Both hospital and medical professionals should consider ascertaining whether patients have this support before hospital discharge. If this support is not available, hospitals can facilitate or provide referrals to social support opportunities for patients. Furthermore, it is known that only 30% of cardiac patients attend rehabilitation; thus, most patients do not access this protective factor to decrease distress [111]. Participation may be increased if patients are educated about the potential benefit for their long-term mental and physical health, including decreased risk of future cardiac events. This review concludes that cardiac patients would also benefit from attending both pre-operative education and cardiac rehabilitation as both are associated with small to large reductions in distress throughout their recovery. Health professionals might provide heavier emphasis on the importance of both of these to their patients’ long-term health. Additionally, due to pre-operative distress predicting post-operative distress, patients should be screened either pre-surgery or pre-discharge and provided with additional psychosocial support if found to be distressed. Finally, health professionals should carefully monitor pain levels in patients pre- and post-discharge as increased pain predicts distress. If patients are found to be in pain, psychosocial support should be provided to them.

## 5. Conclusions

This study reviewed evidence relating to the problem of on post-operative distress among cardiac patients, investigating the role of biographical, psychosocial and health service factors. The evidence suggests that undergoing pre-operative education and having close relationships with family/partner were associated with moderate-to-large reductions in distress. Furthermore, attending cardiac rehabilitation, higher levels of physical activity, and increased social interaction were associated with small to moderate reductions in distress. Moreover, distress post-surgery increased when pain and functional impairment were elevated. Interestingly, age and sex were inconclusive predictors of distress with available evidence attesting that these factors both increased and decreased distress. Pre-existing anxiety and depression had significant impacts on the severity and longevity of distress among cardiac patients. 

In conclusion, participation in pre-operative education appears to be the strongest predictor of distress with strong evidence that close family and partner relationships are also a critical protective factor. Future research should investigate specific elements of cardiac rehabilitation and pre-operative education (e.g., psychosocial management, exercise training, etc.) to identify the most effective components in reducing distress. Additionally, there are several measures that could be employed in future research such as self-efficacy, timing of cardiac rehabilitation, and pre-surgical levels of physical activity. It is clear from the evidence that there are a number of opportunities for health practitioners to actively support reduced patient distress. 

## Figures and Tables

**Figure 1 healthcare-11-02721-f001:**
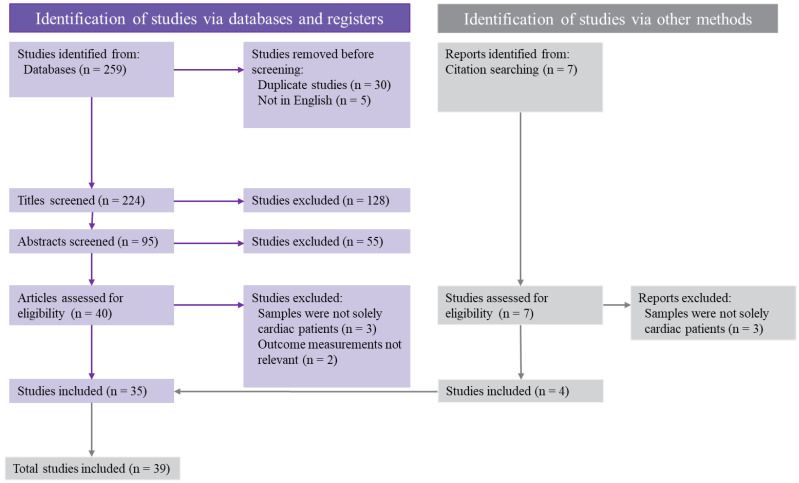
PRISMA diagram of the process used for screening studies.

**Table 1 healthcare-11-02721-t001:** Summary of Predictor Domains Across Trends, Evidence Quantity and Quality.

Predictor Domain	Specific Factor	Depression			Anxiety			Stress			General Distress			Morbidity		
		Trend ^1^	Evidence Qu Ant ^2^	Evidence Qual ^3^	Trend ^1^	Evidence Quant ^2^	Evidence Qual ^3^	Trend ^1^	Evidence Quant ^2^	Evidence Qual ^3^	Trend ^1^	Evidence Quant ^2^	Evidence Qual ^3^	Trend ^1^	Evidence Quant ^2^	Evidence Qual ^3^
Personal	Age	Mixed	Moderate	Excellent	Mixed	Moderate	Excellent									
	Sex	Mixed	Ample	Excellent	Mixed	Ample	Excellent									
	Pain and functional impairment	Small +	Emerging	Good							Small +	Emerging	Good			
	Physical activity	Small −	Ample	Excellent	Small −	Ample	Excellent				Small −	Emerging	Satisfactory			
Interpersonal	Family/partner relationships				Large −	Emerging	Satisfactory	Large −	Emerging	Satisfactory	Small −	Moderate	Good	Small −	Emerging	Satisfactory
	Social interaction/activity	Small −	Emerging	Satisfactory							Moderate −	Moderate	Good	Large −	Emerging	Satisfactory
Health service/system	Pre-operative education	Moderate −	Ample	Excellent	Large −	Ample	Excellent				Small −	Emerging	Good			
	Cardiac Rehabilitation	Small −	Ample	Excellent	Mixed	Ample	Excellent	Small −	Ample	Excellent				Moderate −	Ample	Excellent

Notes: ^1^ Trend describes whether there is a consistent or mixed trend identified in the reviewed evidence. If possible, the trend is then described as weak, moderate, or strong in magnitude and positive (indicated by a + symbol) or negative (indicated by a − symbol) in direction. ^2^ Evidence is rated on its quantity with possible values of emerging (1–5 studies), moderate (6–20 studies), and ample (>20 studies). ^3^ Evidence is rated on quality based on what proportion of studies for each factor are high on the hierarchy of evidence published by the National Health and Medical Research Council where these ratings take values of ‘excellent’, ‘good’, ‘satisfactory’ and ‘poor’.

## Data Availability

No new data were created or analyzed in this study. Data sharing is not applicable to this article.

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
