# Peer review of "Predictors of Psychological Distress among Post-Operative Cardiac Patients: A Narrative Review"

_healthcare, 2023, doi:10.3390/healthcare11202721_

Round 1
Reviewer 1 Report
The manuscript is written in a scientific language. It raises the important topic of psychological aspects of patients after cardiac surgery. The methodology for the literature review is appropriately selected. The authors discuss the limitations of the study. The existing literature was appropriately considered, although the literature covers a very wide range - from 1983 to 2022. I would suggest that authors focus on more recent publications. The clinical implications of the literature review are described, but I would suggest to pay more attention to this part. Please describe how patients should be educated and which preventive decisions should be made.
Reviewer 2 Report
I am grateful that you have entrusted me with reviewing the manuscript. Below, I detail much-needed aspects of improvement. -First of all, the structure of the summary is not very accurate because it loses the reader. It does not emphasize the problem to be analyzed nor does it focus on the problem. It is also necessary to improve the procedural part of the summary. It is necessary to add information about the conclusions part in the summary. - Throughout the manuscript it is necessary to review the writing so that it can conform to what is expected for a scientific manuscript. It is necessary to add and update more bibliographic references since this is a topic of interest. The reader is lost by using quotes that are not recent at all. Authors should take care of this throughout the manuscript. Get recent quotes that bring relevance to the topic and do not forget the importance of citing the journal to which the article is sent. Throughout the introduction it is appropriate to enrich the information with analysis of recent data. This is why it is necessary to review the coherence and structure of the different points discussed in the introduction. Being necessary to use connectors. That is why the hypotheses and objectives must be justified clearly. Therefore, the research questions must be clearly defined and adequately justified in the theoretical framework.
- The methodological framework has certain deficiencies: the inclusion criteria and exclusion criteria used are missing. The procedure must be clarified. - The results part can be improved. The data do not contribute anything with respect to the objectives of the study. The results must be taken into account with respect to the objectives of the study. -In the discussion it is necessary to indicate a greater wealth of previous studies that support or do not support the results provided. - The conclusion is not rich in content because it lacks structure and clarity for the reader. - It is recommended to add some final conclusions of the study that shed light on the results found and that indicate possible lines of research, as well as application. - It is recommended to review the citation regulations.
Reviewer 3 Report
The study does not bring significant novelty to the topic, and its merit is primarily limited to the systematization of already known information. However, the relevance of predictors such as age, sex, pain, and functional impairment, as well as physical activity, remains evident.
As a literature review, what may be lacking is a more in-depth critical analysis and an original synthesis of the available evidence. To achieve this, the following elements are required:
· - A more detailed discussion of the identified relationships: In addition to reporting the associations between predictive factors and post-operative disorders, the authors should attempt to explain the underlying mechanisms behind these associations. This entails a deeper analysis of the theories or hypotheses behind the identified relationships.
· - Consideration of cultural and demographic context: Depending on the reviewed studies, the authors can explore how cultural and demographic factors may influence the relationships between predictors and post-operative disorders.
· - The study can benefit from identifying the practical implications of the reviewed evidence.
· - An original contribution of the review can be to point out new research questions that arise from the identified gaps in the existing literature. This helps guide future research in the field.
Additionally, it is worth noting that the article lacks a concluding section. A well-crafted conclusion could effectively summarize the key findings, highlight the implications of the reviewed evidence, and provide a clear takeaway message for readers.
Round 2
Reviewer 2 Report
I consider that the manuscript in the version has improved.
Reviewer 3 Report
The authors justified the changes made to the manuscript.